# Determination of Cyanotoxins and Prymnesins in Water, Fish Tissue, and Other Matrices: A Review

**DOI:** 10.3390/toxins14030213

**Published:** 2022-03-16

**Authors:** Devi Sundaravadivelu, Toby T. Sanan, Raghuraman Venkatapathy, Heath Mash, Dan Tettenhorst, Lesley DAnglada, Sharon Frey, Avery O. Tatters, James Lazorchak

**Affiliations:** 1Pegasus Inc., U.S. EPA, Cincinnati, OH 45268, USA; sundaravadivelu.devi@epa.gov (D.S.); venkatapathy.raghuraman@epa.gov (R.V.); 2Office of Research and Development, Center for Environmental Solutions and Emergency Response, U.S. EPA, Cincinnati, OH 45268, USA; mash.heath@epa.gov (H.M.); tettenhorst.dan@epa.gov (D.T.); 3Office of Water, Science and Technology, U.S. EPA, Washington, DC 20004, USA; danglada.lesley@epa.gov (L.D.); frey.sharon@epa.gov (S.F.); 4Center for Environmental Measurement and Modeling, U.S. EPA, Gulf Breeze, FL 32561, USA; tatters.avery@epa.gov; 5Center for Environmental Measurement and Modeling, U.S. EPA, Cincinnati, OH 45268, USA

**Keywords:** harmful algal blooms, cyanotoxins, cyanobacteria, fish tissue, shellfish, detection methods

## Abstract

Harmful algal blooms (HABs) and their toxins are a significant and continuing threat to aquatic life in freshwater, estuarine, and coastal water ecosystems. Scientific understanding of the impacts of HABs on aquatic ecosystems has been hampered, in part, by limitations in the methodologies to measure cyanotoxins in complex matrices. This literature review discusses the methodologies currently used to measure the most commonly found freshwater cyanotoxins and prymnesins in various matrices and to assess their advantages and limitations. Identifying and quantifying cyanotoxins in surface waters, fish tissue, organs, and other matrices are crucial for risk assessment and for ensuring quality of food and water for consumption and recreational uses. This paper also summarizes currently available tissue extraction, preparation, and detection methods mentioned in previous studies that have quantified toxins in complex matrices. The structural diversity and complexity of many cyanobacterial and algal metabolites further impede accurate quantitation and structural confirmation for various cyanotoxins. Liquid chromatography–triple quadrupole mass spectrometer (LC–MS/MS) to enhance the sensitivity and selectivity of toxin analysis has become an essential tool for cyanotoxin detection and can potentially be used for the concurrent analysis of multiple toxins.

## 1. Introduction

Harmful algal blooms (HABs) have been observed in freshwater, estuarine, and marine waters in the U.S. and around the globe. Cyanobacteria frequently contribute to HABs in freshwater systems and are able to produce highly potent toxins, known as cyanotoxins. A large number of cyanotoxins have been reported from different species of cyanobacteria, including microcystins (*Microcystis*, *Anabaena*, *Hapalosiphon*, *Dolichospermum*, *Gloeotrichia*, *Nostoc*, *Oscillatoria*, *Phormidium*/*Microcoleus*, and *Synechocystis*), nodularins (*Nodularia*, *Nostoc*, and *Iningainema*), anatoxins (*Anabaena*, *Aphanizomenon*, *Cylindrospermum*, *Dolichospermum Planktothrix*, *Oscillatoria*, *Geitlerinema*, *Phormidium*/*Microcoleus*, and *Tychonema*), cylindrospermopsins (*Oscillatoria* and *Raphidiopsis*), lyngbyatoxin-a (*Lyngbya*), saxitoxins (*Anabaena*, *Aphanizomenon*, *Cylindrospermum*, *Lyngbya*, *Phormidium*, and *Raphidiopsis*), and aplysiatoxins (*Lyngbya*, *Schizothrix*, and *Oscillatoria*) [1,2,3,4,5,6,7,8,9,10,11].

Microcystins (MCs) are the most frequently studied and the most widespread cyanotoxins [12], with approximately 50% of publications on cyanotoxins focusing on MCs, 25% on saxitoxins, and 25% on other toxins (such as nodularin, anatoxin-a, β-N-methylamino-L-alanine, cylindrospermopsins, and prymnesins). Of these publications, a majority focus on detection in water matrices. There is limited information on the presence of MCs and other cyanotoxins in matrices such as aquatic food webs, phytoplankton, zooplankton, periphyton, macroinvertebrates, forage fish, bottom feeders, and top carnivore fish. Since research on cyanotoxins has mostly focused on MCs, especially in water matrices, the development and standardization of better monitoring methods for other cyanotoxins and complex matrices, such as those mentioned above, are of utmost importance.

Various techniques have been developed to analyze cyanotoxins in water. These methods include enzyme-linked immunosorbent assay (ELISA), protein phosphatase inhibition assay (PPIA), oxidation of MCs to produce erythro-2-methyl-3-methoxy-4-phenylbutyric acid (MMPB) [13], and chromatography coupled with various detection methods [14]. A survey of chromatographic methods for the detection of cyanotoxins in water could be divided into the following main categories: liquid chromatography with ultraviolet detection (LC–UV), LC with fluorescence detection (LC–FLD), LC–photo-diode array (LC–PDA) or similar, LC with tandem mass spectrometry (LC–MS/MS), gas chromatography with mass spectrometry (GC–MS), ion mobility spectrometry (IMS), and immunological assays. LC–MS, or single quadrupole mass spectrometry, has largely been replaced by LC–MS/MS. Among these methods, a few LC–MS/MS and ELISA-based standard methods developed by the U.S. Environmental Protection Agency (EPA) and the International Organization for Standardization (ISO) are available, are considered standard, are widely used for the measurement of cyanotoxins in water, and can potentially be adapted for use with other matrices, such as fish tissue [15,16,17,18]. However, there is no standardized analytical method to detect these cyanotoxins in other matrices [19]. The surveyed studies herein show that ELISA and LC–MS/MS analysis have most frequently been used for detection/quantitation. While specific extraction procedures used vary considerably, overall method performance metrics, such as matrix spike recoveries and replicate precision, were sometimes poor or unreported. This raises questions about the quality of the data in the case of fish tissue and organs, shellfish and similar matrices exposed to HABs.

Most methods developed to date depend on extracting toxins in some form of an aqueous matrix, typically water and one or a combination of organic solvents. For example, ELISA methods should be carried out directly in water with only limited amounts of organic solvent, while LC–MS/MS methods typically require samples to be prepared with significant amounts of organic solvent in water [20]. In addition to the presence of organic solvents, some methods are sensitive to low pH, lipid content, or other matrix effects due to the properties of the target toxins or the nature of the matrix [21]. Hence, analytical methods for cyanotoxins in complex matrices, such as tissue, organs, and plants, frequently require efficient extraction, clean-up, and transfer of the toxin into a suitable solvent for analysis. Use of inefficient extraction methods may lead to poor matrix spike recoveries and poor precision among replicate samples.

The prevalence of HAB events globally and the potential for exposure and bioaccumulation of toxins necessitate monitoring and detecting cyanotoxins in tissues associated with HAB-related mortality events. There is a need for the development of reliable extraction and analytical methods capable of analyzing edible fish/shellfish, plant, and animal tissues for multiple toxins and for such methods to be broadly applicable to all inland waters and coastal food webs. The overarching goal of this study is to summarize available methodologies for the measurement of toxins, identify gaps in the detection of toxins in complex matrices, and highlight the complexity of quantifying groups of compounds with diverse chemistries.

## 2. Cyanotoxins and Prymnesins

Cyanotoxins can be categorized based on (1) their mechanism of action on terrestrial vertebrates, especially mammals, e.g., hepatotoxins, neurotoxins, and dermatotoxins, and (2) their chemical structure, e.g., cyclic peptides, alkaloids, and lipopolysaccharides [22,23,24]. Table 1 lists the names, chemical groups, and mammalian targets of some common cyanotoxins. Physicochemical properties of cyanotoxins not only determine which extraction procedures or detection techniques are suitable but also impact the bioaccumulative and toxicological properties of the toxins on several taxa of aquatic invertebrates and vertebrates [25].

MCs, the most studied class of cyanotoxin, are produced several cyanobacterial genera including *Microcystis*, *Anabaena*, *Hapalosiphon*, *Nostoc*, *Dolichospermum*, *Gleootrichia*, and *Oscillatoria* [12]. These compounds are short cyclic heptapeptides synthesized by non-ribosomal pathways [26]. MCs are characterized by significant structural variability in amino acid composition, including residue substitutions, methylations, and demethylations. In total, more than 248 congeners/variants have been reported in the environment. Among them, MC-LR and MC-LA are among the most toxic [27,28]. Only a limited subset of MCs has commercially available analytical standards, which limits the scope of quantitative methods, such as LC–MS/MS, which rely on these for accurate quantification. MCs are cytotoxic and inhibit protein phosphatases, which leads to several subsequent harmful effects at concentrations over 10 µg/L [29].

Nodularins (NODs) are a class of cyclic pentapeptides structurally similar to MCs. Currently, 10 variants of this water-soluble and stable toxin have been identified [30,31], the most common being the variant with arginine as the variable amino acid (NOD-R). Similar to MCs, NODs are hepatotoxins acting through the inhibition of protein phosphatases and are potential tumor promoters. While NODs have historically been considered to be less common than MCs, their occurrence has been observed in bloom events corresponding to the species *Nodularia spumigena*, and animal deaths have been reported [31].

Anatoxin-a (ANA-a) and its structural variants have been associated with the genus *Anabaena* (now *Anabaena*/*Dolichospermum*) and also to *Aphanizomenon*; *Planktothrix*; *Cylindrospermum*; *Microcystis*; and the benthic *Oscillatoria*, *Phormidium*, *Tychonema*, and *Geitlerinema* [32]. ANA-a is an alkaloid known to exhibit acetylcholinesterase inhibition activity, which is its primary mode of toxicity. Of the structural variants of ANA-a, homoanatoxin-a, resulting from the methylation of the carbon at the extremity of the ketone function, is one of the more commonly observed [33]. Several other derivatives of ANA-a have been identified, including 2,3-epoxy-anatoxin-a; 4-hydroxy- and 4-oxo-derivatives; dihydroanatoxin-a; and dihydrohomoanatoxin-a [34,35]. Toxicity indications for these other variants are mixed, with some reports indicating lower toxicity while others suggesting higher oral toxicity from the dihydroanatoxin variant specifically, suggesting that monitoring of these variants is of interest [36].

Saxitoxins (STXs) are highly polar, nonvolatile, tricyclic perhydropurine alkaloids. While the overall molecular structure is largely conserved within the group, substitutions and variations amount toat least 57 analogs in the environment. They are also referred to as paralytic shellfish toxins (PSTs). PSTs can be non-sulfated, singly sulfated, or doubly sulfated [33,37]. These water-soluble toxins can persist for over 90 days in freshwater [38], but they are altered by high temperatures and may be degraded into more toxic variants [32]. STX is one of the most potent natural neurotoxins, and a dose of approximately 1 mg of the toxin from a single serving of contaminated shellfish is fatal to humans [39].

The cyanotoxin β-N-methylamino-L-alanine (BMAA) is a non-protein amino acid reportedly produced by the majority of cyanobacterial isolates [40]. Some literature reports have suggested that cellular exposure to BMAA may lead to neurological damage in the brain and the central nervous system of humans and animals, potentially contributing to one of several neurodegenerative diseases [41]. However, this remains uncertain as while other studies have illustrated a wide spectrum of effects to exposure, the neurodegenerative disease relationship was not universally observed [42].

Cylindrospermopsins (CYNs) are another class of alkaloid cyanotoxins produced by a variety of freshwater cyanobacteria. Production of CYNs have been reported from Nostocalean species mostly, as well as recently from one Oscillatoriale [43]. CYN and its variants are highly polar, polycyclic uracil derivatives containing guanidino and sulfate groups, and their detection has been reported worldwide. It is a potent hepatotoxin and is likely to be taken up by a variety of aquatic organisms, suggesting potential bioaccumulation risks [22,44].

The invasive algae *Prymnesium parvum* (golden algae) is a species of haptophyte (Prymnesiophyta). The species is of concern because of its ability to produce a suite of complex polyether toxins, known as prymnesins. Prymnesins may refer to Prymnesin-1, Prymnesin-2, or Prymnesin-B1 (to name a few of the presently identified species). These variants differ primarily in the length of the carbon backbone and are recognized hemolytic and ichthyotoxic agents that have been associated with massive fish kills in at least 14 countries [45]. Because of the uncertainty as to the identity of potentially toxic species, and the unavailability of analytical standards, the detection methods used for other cyanotoxins are often not suitable for the detection of prymnesins, which often require more complex analyses, such as high-resolution mass spectrometry (HRMS), quadrupole-time of flight mass spectrometry (qTOF), and nuclear magnetic resonance (NMR) [46,47].

## 3. Current Detection Methods for Cyanobacterial Toxins

A wide range of detection methods is available for the analysis of cyanotoxins. The most commonly used methods for different toxins are listed in Table 2. These methods are primarily targeted for water matrices; however, some can potentially be adapted for use with other matrices, such as fish tissue, shellfish, or organs.

The quantification of cyanotoxins in water matrices is at this point a well-studied problem, with multiple EPA standard methods developed for drinking water and an extensive literature presence. However, studies on the extraction and quantification of cyanotoxins in more complex matrices, such as shellfish, fish tissue, and organs, are more limited in quantity and scope. This review will present a summary of studies evaluating cyanotoxins in various matrices, and their extraction, preconcentration, and clean-up stages will be discussed, along with the specific quantitation techniques employed.

## 4. Sample Preparation and Analytical Methods for Cyanotoxins and Prymnesins Detection

Various methods are available for the quantification of cyanotoxins, as described above, and are primarily split between immunoassays and chromatographic separation (i.e., GC or LC), followed by quantification using mass spectrometry, UV, and photodiode array detectors. The matrix suitability varies by procedure, with a variety of interferences inherent to specific analysis; for example, mass spectrometric methodologies are susceptible to matrix interferences related to the ionization technique employed, with ion suppression/enhancement commonly observed in more challenging sample matrices. For each method, then, the samples often require specific preparation before analysis. This is especially common for tissue and other more complex matrices than water.

### 4.1. Sample Preparation

Sample preparation procedures differ according to the toxin analyzed and the nature of the tissue, organ, or biomass analyzed. Typically, this includes extraction, clean-up, and any post-extraction modifications (e.g., derivatization and solvent exchange) prior to analysis. Table 3 lists a variety of sample extraction and preparation procedures from a survey of the literature. The typical workflow being followed is summarized in Figure 1.

Procedures for extracting toxins from complex matrices vary widely (Table 3). In most cases, heterogeneous samples, including tissues, organs, and food supplements, are mechanically homogenized or lyophilized to break down any solid structures and produce a suspension or emulsion amenable to extraction by solvents containing various percentages of aqueous and/or organic solvents (acidified methanol, acidified water, acetone, acetonitrile, butanol, etc.). In samples containing intact cyanobacterial cells, lysis to release intracellular toxins is often facilitated by ultrasonication, extended mixing, or incubation at a specific temperature and application of heat [48]. Following lysis/homogenization and extraction, a variety of clean-up techniques are available (Figure 1), including centrifugation, solid-phase extraction (SPE), protein precipitation, hexane washes, filtration, and immunoaffinity columns [49]. The clean-up steps attempt to remove matrix components, after which the toxin is eluted/extracted with a solvent and reduced in volume by evaporation prior to analysis. The efficiency of extraction techniques and the clean-up steps in the recovery of cyanotoxins from complex matrices vary widely based on the toxin physicochemical properties, matrix effects, holding times, collection methods, and other site-specific variables. In addition, each of these extraction techniques and clean-up steps may result in some loss of the target toxins during sample processing.

Extraction of cyanotoxins from complex matrices requires use of appropriate solvent mixtures to remove them, ideally without also extracting interferences to the analytical techniques being employed. The references cited in Table 3 are, broadly, either attempting to extract only individual types of toxins or attempting to quantify multiple classes with a single extraction workflow. The former can potentially allow for optimal conditions to be obtained, while in the case of more general extractions, compromises may result from attempting to measure chemicals less compatible with the extraction conditions. The primary differentiating step is the extraction procedure, solvents, and conditions used, which are elaborated upon below.

#### 4.1.1. MCs and NODs

There are numerous literature examples of procedures for the extraction of MCs from a variety of matrices, such as tissue, listed in Table 3, including some that have systematically evaluated various extraction conditions. Because of their similar structure and chemical properties to MCs, there is little to differentiate extraction conditions for NODs, and as seen in Table 3, the conditions are comparable. The most common methods for the extraction of MCs have used methanol:water mixtures of varying ratios, in some cases with added acid, typically acetic acid. Because MCs are a diverse class of species with varied amino acid substitutions, one complicating factor is the variability in hydrophobicity/hydrophilicity of MCs that could be present in a sample. This has been found to be an issue in extractions with pure methanol, for which lower recovery of more hydrophilic congeners, such as MC-RR, has been reported. Instead, most procedures have used 75% to 90% methanol [50], the remainder being water, which has been seen to be generally appropriate for a broad suite from hydrophobic to hydrophilic congeners (See Table 3). An additional complication, however, has been observed in extractions using methanol when used in conjunction with some ELISA based toxin measurements. In at least some studies, interferences resulting in false positive detections were observed and confirmed by comparison with LC–MS/MS, where no MCs were observed [49]. While it is possible that the targeted methods were missing unknown MCs, this was further evaluated by comparison with a direct monoclonal ELISA with known higher specificity to MCs, and the absence of MCs via that method was confirmed.

One alternative to attempting to extract individual MC congeners has been developed using Lemieux oxidation, which has been demonstrated to react with a variety of MCs and convert them to a common product from all MCs with “adda”, producing MMPB, which can be extracted instead. Quantification of MMPB can then serve as a proxy measurement for the total of MC congeners present in the sample. This does have limitations in that MMPB oxidation is known to generate positive results when parent MCs may be partially metabolized or otherwise transformed, as in drinking water treatment where oxidized MCs give a false positive signal, but recent publications indicate correlations seen between solution phase MC detections and detections in fish tissue [51].

#### 4.1.2. ANA-a

Examples in Table 3 show three different procedures for the extraction of ANA-a from a solution. The first uses an acidic ethanol:water mixture (80:20) that is largely comparable to that for the majority of MC detecting methods above [52]. An alternative approach used immunoaffinity beads to pull the toxin from waters and slurries, but the practicality of this for heterogeneous systems was less clear [53]. Finally, for water samples, SPE is a viable avenue for extracting ANA-a, but this is again of limited utility for more heterogeneous matrices, where cartridge clogging will be more of a challenge [54]. More recent examples for ANA-a extraction in mixed methods are discussed below.

#### 4.1.3. STXs

STXs are universally prepared using acidic or buffered extractions owing to their chemical instability under basic conditions. As these are a hydrophilic class of toxins, the extraction solvents used have generally been acidic, typically aqueous acetic [55,56,57] or hydrochloric acid [58], with buffered water being used in one instance [59]. Elevated temperatures are frequently used to extract STXs, including boiling of the solvent/sample mixture to facilitate extraction efficacy. Interestingly, in one study [59], acidic water and acidic methanol:water mixtures were evaluated and found to produce significantly lower recoveries of STX specifically, while the use of neutral pH phosphate buffer increased recoveries to >50%, which they speculated was due to reduced solubility in low pH.

#### 4.1.4. CYNs

Cylindrospermopsins are a significantly more hydrophilic toxin than MCs, without as much structural diversity, and as a result extraction procedures have generally been more straightforward. In Table 3, three procedures are listed for the analysis of CYNs, and these all use mechanical homogenization or lyophilization, followed by extraction with methanol or water.

#### 4.1.5. BMAA

Extraction of BMAA is typically performed under acidic conditions following lyophilization [60,61,62]. Because it is an amino acid, it is highly water soluble and does not require the use of organic solvents for extraction from tissue. However, removal of proteins and/or lipids is part of the process in some cases on extraction with chloroform [61].

#### 4.1.6. Prymnesins

Evaluation of extraction techniques for prymnesins and other toxins associated with *P. parvum* is difficult owing to the many questions related to the toxic species, their structure, and the lack of available analytical standards. The protocols listed in Table 3 illustrate these challenges, with [46,47,63] repeatedly re-extracting the water/algal cell lysates with solvents of varying polarity in an attempt to isolate and characterize potential toxic species. In [46], cold acetone was used as a pre-extraction solvent to remove chlorophyll, followed by methanol extraction, in which prymnesins were observed by LC–HRMS. In [47], *P. parvum* lysates were extracted with dichloromethane, ethyl acetate, methanol, and water and while prymnesins were not observed under LC–HRMS analysis, cytotoxic activity was seen in the ethyl acetate and methanol extracts and a variety of fatty acid amides and one hydroxamic acid were observed. Finally, in [63], cold acetone was again used as a chlorophyll-removal step before sequential methanol and *n*-propanol extraction of the cellular material, which was then followed by solvent exchange to water. The aqueous material was then extracted with ethyl acetate four times to remove fatty material, before a final SPE step for clean-up prior to analysis. In this study, two prymnesins and a variety of related ions and fragments were characterized by LC–HRMS.

#### 4.1.7. Extraction of Multiple Cyanotoxins

Workflows to extract multiple toxin classes typically attempt either to group compatible species together or to accept compromises in recovery arising from the impossibility of trying to recover diverse species simultaneously. As described above, procedures for the extraction of MCs and NODs have typically settled on 75–90% methanolic water as an extraction solvent. In studies such as [20,64,65,66], extraction of both MCs and NODs is described and the solvent mixtures used were uniformly in that range, showing the ease of extracting similar classes of toxins in tandem. However, studies attempting to extract those toxins as well as the more hydrophilic species, such as STXs or CYNs, as in [67,68,69,70], typically reduced the percentage of organic solvent to ~50% in order to improve solubility of the more hydrophilic constituents of the methods. An alternative approach was followed in [71], where two separate extraction procedures were devised to recover collectively STX, ANA-a, and CYN with 25% acetonitrile in water, while MCs and NOD were extracted with 75% acetonitrile in water from a split sample. The latter approach attempts to reduce the impact of chemical incompatibility, but at a cost of doubling extraction and analysis requirements. Once the samples are prepared based on the extraction and clean-up techniques described above, the many analytical methods discussed in Section 4.2 may be used for detection and quantification.

### 4.2. Analytical Methods

#### 4.2.1. Immunological Assays

Cyanotoxins can be detected through recognition and binding to specific antibodies, either monoclonal or polyclonal. For example, ELISA kits are commercially available for the detection of MCs in water based upon either the common “adda” moiety present or specific recognition of a single MC congener [49,72,73,74]. Depending upon the antibody and the procedure employed, these kits can achieve a detection limit (DLs) as low as 4 ng/L, with an upper quantitation limit (due to saturation) of 5 μg/L for MC-LR [73]. While ELISA is frequently employed for the detection of MCs, particularly in drinking water, ELISA kits have also been made commercially available for the detection of ANA-a, CYN, and STX [75,76]. The most significant advantage of ELISA methods is that they do not require expensive and high-upkeep analytical instrumentation to be maintained, as they typically rely only on colorimetric assays for quantification.

However, detection methods based on ELISA have some limitations of varying severity by target compound. The measurement of a variety of MCs by ELISA is possible because the antibody assay is broadly cross-reactive over different MC congeners; however, this cross-reactivity is not uniform, and in some cases, MCs may be measured with greater or lesser responses relative to MC-LR [20]. Beyond that, only a single signal for a general MC concentration will be obtained, even for a sample that might contain a variety of MC congeners of varying potential toxicity levels. As a result, ELISA-based measurements should be considered to be semi-quantitative in that the measured concentrations are influenced by a number of variables that may or may not be known at the time of analysis. In addition to this limitation, cross-reactivity of the assay with other compounds in the sample and matrix may lead to over- or underestimation of the concentration of toxins, for both MCs and other cyanotoxins. This was observed in one paper, mentioned in Table 3, where ELISA kits targeting the “adda” moiety common to most MCs were found to be generating false positive results when compared with LC–MS/MS and an alternative, monoclonal antibody specific to MC-LR [77]. Conversely, it has also been demonstrated that ELISA-based screening may fail to measure protein-bound or glutathione-conjugated species in tissue matrices, perhaps due to the molecules being inaccessible to the antibody. When ELISA extracts were compared with a chemical oxidation/derivatization technique for measuring total MCs in tissue, a significant enhancement in the measured concentrations was observed in the latter [78]. These types of issues are not specific only to ELISA, but rather a general complication for any measurement technique. The sequestration/transformation of toxins through metabolic/removal processes may make them unavailable for many types of assays.

ELISA-based techniques can also be sensitive to the presence of solvents such as methanol and acetonitrile that are often used for tissue extraction. Commercially available ELISA kits typically recommend <5–10% solvent, but this value may be as low as <2.5% for ANA-a and as high as <20% for CYN and STX, based on standard protocols included in the various ELISA kits; therefore, in many studies relying on ELISA to quantify toxins from tissue extracts (see Table 3), the organic extracts are evaporated to dryness under nitrogen at 30–60 °C and reconstituted in DI water or an appropriate diluent prior to analysis [20]. Low pH can also affect ELISA performance, which could be an issue for STX extracted under acidic conditions. In many of the reviewed studies, pH was adjusted with 0.1 N NaOH to improve compatibility with the ELISA assay.

#### 4.2.2. Mass Spectrometry

Following sample extraction and clean-up, separation of compounds typically employs either liquid or gas chromatographic (LC or GC) techniques. Typically, LC methods use a reversed phase C18 or a hydrophilic interaction liquid chromatography (HILIC) column and either methanol:water or water:acetonitrile gradients for separation, as these allow for flexibility, speed, and adaptability to a wide range of detectors relying on UV absorbance, fluorescence, or mass spectrometry. GC can be used as a separation method for cyanotoxins [79]. However, many cyanotoxins, including MCs, are larger molecules and are either not volatile or do not ionize well without chemical derivatization techniques. In addition, GC-based methods might require a solvent-exchange step to a nonpolar solvent such as hexane or ethyl acetate prior to analysis, which also typically involves blowing samples down to dryness prior to reconstitution with the nonpolar solvent. As such, GC-based separation requires more complex and time-consuming sample preparation, while also presenting more avenues for toxin loss from samples. While there are examples in the literature of analytical methods for cyanotoxins using GC–MS methods, these are a clear minority [78]. The single example in Table 3 using GC–MS involved MMPB oxidation of MCs, and in this instance, using GC–MS as the detector required an added derivatization step in the workflow.

Mass spectrometric methods for quantifying cyanotoxins rely on comparison with specific analytical standards for target compounds and can provide greater specificity than is possible with ELISA. In the LC–MS/MS techniques typically employed for toxin measurement, both a starting mass and a characteristic fragment ion specific to a given chemical are used to ensure specificity of assignment, along in some cases with a separate confirming ion fragment and a unique ratio of confirming to main ion. This is advantageous in that it means that false positive identifications are less common (although not impossible; ANA-a is isobaric with the common amino acid phenylalanine and both parent and product masses are identical; confusion of these two species can occur when MS-based detection is used [53]).

An additional useful feature of LC–MS/MS methods is that surrogate and internal standards, usually consisting of isotopically labeled analogs of target compounds, can be added to samples prior to extraction and analysis. This enables the losses encountered during extraction, clean-up, and analysis steps to be accounted for through comparison with expected recoveries of surrogates, while isotopic dilution techniques can allow for compensation for some matrix interferences during all of these processes, improving overall quantitation accuracy [77].

Quantitative mass spectrometric methods have been developed for the majority of the cyanobacterial toxins, including MCs, ANA-a, CYNs, STXs, and BMAA, either as a native compound or after chemical modification [80]. This allows samples to be separated using conventional C18 phases, although with ANA-a(S), HILIC methods work well for the assessment of this very hydrophilic toxin [81]. Due to the specificity, sensitivity, and rapidity of the analysis, mass spectrometric methods are now the physicochemical method of choice for the quantitative analysis of most cyanobacterial toxins in complex matrices for labs with access to appropriate instrumentation.

Cyanotoxins can also be detected by MS without preliminary chromatographic separation, particularly with time of flight (TOF) mass spectrometers in which many compounds can be identified and quantified concurrently. For example, matrix-assisted laser desorption/ionization coupled with TOF (MALDI-TOF) can be used to perform toxin analysis in even individual cell colonies [79]. In a typical workflow, target compounds enclosed in the dried and solid sample are ionized by a laser beam and accurately identified through the high mass resolution provided by the TOF instrument. However, TOF mass spectrometers usually tend to be less sensitive than other mass spectrometers of the same generation, and these methods are less commonly used for routine sample screening and quantitation than LC–MS/MS methods. However, TOF- and HRMS-based instruments can be used to qualitatively identify toxins that lack analytical standards, unknown toxins, and/or their metabolites or quantify using standards of other structurally similar toxins [82,83,84].

LC–MS/MS is the most commonly used chromatographic method for cyanotoxin detection in general, but it is limited by the need for analytical standard material to be available. For example, LC–MS/MS methods for the measurement of prymnesins and other toxins produced by *P. parvum* have not been commonly reported in the literature; these represent an emerging class of contaminants for which there is presently little information. The key issue preventing their measurement is the absence of a pure analytical reference standard for these toxins, which could be used for calibration. The ability to incorporate an internal standard would be ideal for highest precision for an analytical method, ideally a stable, isotopically labeled form of the analyte, but along with the native (unlabeled) prymnesins, these are not available. To date, only a few studies have reported isolating potential toxins, including prymnesin molecules, from *P. parvum*, [85,86,87,88,89], and replication of this work is yet to be reported in the peer-reviewed literature. Analytical standards for the prymnesins, either normal or isotopically labeled, along with any other structurally similar molecule are not currently available. A few studies have used UTEX strain 2797 as the source material for the preparative isolation of toxin material for analytical method development [46,63,89,90,91,92], but these preparations are not quantitatively exact enough to allow for use as true analytical standards. As a result, there are currently no published validated methods for the quantitative analysis of prymnesins [93].

#### 4.2.3. LC–UV and LC–FLD

Monitoring UV absorbance was historically one of the first techniques to detect cyanotoxins after LC separation. MCs and CYNs have specific UV spectra with maximum absorbances at 240 and 262 nm, respectively [94,95]. Analytical workflows using LC–UV and/or LC–PDA allow for measuring the concentration of MCs by using these characteristic absorbances. In conjunction with good chromatographic separation, samples can not only have toxin concentrations measured but also give some information on the congeners of the MCs present. The key limitation of UV-absorbance-based techniques, however, is that the absorbance is not limited to only MCs and that background interferences may also impact the baseline signal. This can result in both lower DLs and potentially false positive signals, depending on the matrices being studied. Literature results for MC monitoring using LC–UV and LC–PDA have allowed for measurement of up to seven MC variants through comparison with analytical standards [96,97].

Detection by fluorescence is also commonly used after LC separation. However, cyanotoxins do not naturally fluoresce and, therefore, require the addition of a derivatization process during the sample preparation. For example, in the high-performance liquid chromatography with fluorescence detection (HPLC–FLD) system used by [98], post-column derivatization was performed using a solution containing 10 mmol/L of periodic acid and 550 mmol/L of ammonia in water (flow rate 0.3 mL/min). Nitric acid (0.75 mol/L in water; flow rate 0.4 mL/min) was used for reducing the pH value to 2–3. Fluorescence detection was applied for the determination of STX oxidation products at an excitation and emission wavelength of 330/395 nm.

While most studies that have used LC–UV and LC–FLD techniques have done so in water matrices, several prior studies have used fluorescence and/or UV-based methods to measure cyanotoxins (and in particular MCs) in fish, sediments, and plants [96,99]. The results of these studies show sensitivities comparable to those of LC–MS/MS methods, but with reduced specificity, due to which they can be more susceptible to matrix interferences, resulting in higher practical DLs. While LC–UV, LC–FLD, and LC–PDA methods can be simple and cost effective, misidentification can occur due to the non-specific nature of these methods. LC–MS/MS methods are becoming more prevalent in environmental laboratories now and are the preferred method for toxin identification as they can precisely and accurately identify toxins based on a specific mass/charge precursor ion that is unique to each toxin. In addition, despite typically offering only a unit mass resolution, LC–MS/MS methods can improve confidence in the identification of target compounds through screening for confirmation ions and the ratio of target and confirmation ions being produced.

#### 4.2.4. Biochemical Assays

MCs and NODs are potent inhibitors of protein phosphatases (PPs), and in addition to antibody screening, these toxins can also be detected and measured using a protein phosphatase inhibition assay (PPIA) [74,100,101,102,103]. This assay measures the rate of formation of *p*-nitrophenol (pNP; yellow color) through hydrolysis of *p*-nitrophenol phosphate by PPs over time and measures all PP inhibitors present in a sample. The colorimetric PPIA has been optimized for MC and NOD detection in cyanobacteria extracts using 96-well plates and has shown acceptable correlation with HPLC data [104]. However, PPIA cannot distinguish co-occurring variants of MCs and cannot distinguish MCs from NODs. Therefore, results are often expressed as equivalent dosages relative to MC-LR, which is used as a reference standard. In addition, when analyzing bloom-containing water, interferences with unknown compounds can occur, leading to overestimation or underestimation of toxin concentration. Few studies have documented the use of PPIA for toxin measurement in tissue because of the complexity of tissue extracts and potential for interferences [64,105]. Note that because PPIA detects only inhibitors of that enzyme, within the realm of cyanotoxins it is known to quantify only MCs and NODs and alternative analysis should be undertaken to detect other cyanotoxins if required.

## 5. Discussion

Methods for the extraction and measurement of toxins from complex matrices are influenced by the chemistry of both the toxins and the composition of various matrices. MCs are a particularly diverse class of cyanotoxins, with more than 248 structurally diverse congeners identified in the environment, with chemical properties (including hydrophobicity) varying accordingly. Analysis of MCs in tissue and organs primarily relies on the extraction of homogenized tissue with various fractions of methanol:water, typically 75–80%, or conversion of MCs to MMPB to enable measurements of a single species rather than an ensemble [78,106,107,108]. In both pathways, the extracts are often cleaned via centrifugation [106], hexane wash [108], or SPE [107]. The methanol:water extracts are compatible with LC–MS/MS or other chromatographic methods directly or with ELISA analysis after dilution or solvent evaporation and reconstitution (to reduce the organics in the final sample) to avoid matrix effects [109]. While only a limited number of individual congeners can be quantified using LC–MS/MS methods due to the need for matching analytical standards (in limited availability) and difficulty with isomer separation, both ELISA and the MMPB method can be used to measure total MCs, with the latter potentially also including bound, conjugated, or partially degraded MCs. DLs for ELISA and LC–MS/MS methods for tissue matrices in the literature are generally as low as a few nanograms/gram. DLs for PPIA [106] in water matrices were higher, on the order of 1 µg/L in water. LC–PDA methods also generally have higher detections for MCs in water, with 2.9 µg/L as the lowest level detected, in [97]. Total MCs quantified using the MMPB method, as a proxy for total MCs, have been shown to have DLs of 2.18 ng/g wet weight [51]. One concern for anti-adda-based ELISA methods in the measurement of MCs is the potential for false positives, discussed above; confirmation of ELISA results with LC–MS/MS or direct monoclonal antibodies has been found to be effective at confirming or refuting such results [49].

Due to their structural similarity to MCs, NODs are also frequently extracted using methanol or methanol:water mixtures. If only pure methanol is used, as in [110], if employed for a simultaneous analysis for MCs, it would be expected to be susceptible to the same under-extraction of more hydrophilic MCs, as seen in [72]. More broadly, there is cross-reactivity in ELISA analysis between NODs and MCs, which can lead to difficulties differentiating the specific toxins at a site. LC–MS/MS methods should be used to confirm the specific toxins present if this information is needed. In the reports discussed herein, detection of NODs in tissue varied significantly from organ to organ, from 21 µg/kg to 1.4 mg/kg, with ELISA being the predominant detection method used for tissue samples.

In this review, methods for the analysis of solely ANA-a or CYN were limited and some are more than a decade old. These studies employed extraction of ANA-a using solvents (ethanol:acetic acid, 80:20), immunoaffinity beads, or SPE, followed by analysis using ELISA, IMS, or HPLC–UV [52,53,54]. While DLs for these methods were comparable to extraction/analysis procedures reviewed for other toxins (25 ng/L in water and 4 µg/g in tissue), they are rarely used. Similarly, the extraction of only CYN was achieved using methanol or water, followed by analysis using ELISA, LC–MS/MS, or LC–UV methods, achieving detection varying from 2.7 ng/g for the highly sensitive MS/MS methods to 2.5 µg/g using LC–UV [111,112,113]. LC–MS/MS is the most common methodology currently employed to detect ANA-a and CYN concurrently in the studies reviewed. Since most recent studies optimize the extraction of multiple toxins in one workflow, the multi-toxin methods reviewed herein provide the path forward and best practices for ANA-a and CYN detection in complex matrices. For example, the use of 75:25 water:acetonitrile for the extraction and use of chemically similar isotopically labeled surrogates, such as ANA-a-^13^C_4_ or CYN-^15^N_5_, which have similar extraction recovery, column retention, and ionization efficiency, can help with improved compensation for recovery bias and matrix effects [71,114]. These highly sensitive methods provide method detection limits (MDLs) as low as 0.14 ng/g for ANA-a and 0.12 ng/g for CYN [71].

The methods for the analysis of only BMAA in tissue were also limited in scope. Because it is a small amino acid and highly hydrophilic, studies that concurrently analyze BMAA with these other toxins are limited because of the need for either chemical derivatization or specialized HILIC LC–MS/MS methods, as seen in [60,61,62]. In general, the extraction of BMAA from tissue has been accomplished with an acidic solvent, but the need for hydrolysis, derivatization, and clean-up steps to reduce matrix interferences have made the process lengthy and potentially incompatible with other toxins. Following extraction, analysis was performed using HPLC–FLD or LC–MS/MS, with measurements ranging from micrograms/gram to milligrams/gram. The data from both analysis methods are generally comparable, indicating that the use of HPLC–FLD could be a beneficial first step or screening method. Studies support the use of LC–MS/MS as a necessary confirmation tool [115,116].

Most methods for STX detection in complex matrices usually involve extraction under acidic conditions (with hydrochloric acid or acetic acid, for instance) and often at an elevated temperature [55,56,57,58]. Detection was accomplished with ELISA, HPLC–FLD, LC–MS/MS, or LC–qTOF MS, with DLs mostly in the micrograms/kilogram range. The ELISA method is sensitive and allows for rapid screening of a large number of samples, and LC–MS/MS serves as a useful confirmation tool. STX is stable at high temperatures, which is an important consideration for food safety; it also plays a role in many extraction procedures that require the sample to be boiled [55,56,57,58,59,117]. STX is known to persist for over 90 days and is considered extremely stable even at a high temperature and low pH [38]. Although it can be stored in acid without loss for many decades, some studies suggest that it may not be stable at pH > 8, even at ambient temperature [118]. This suggests that STX may not be stable in water and bivalves unless it is stabilized with acid. The stability also depends on the chemical structure of specific compounds; GTX1, GTX4, and NEO variants are less stable at acidic pH than GTX2, GTX3, and STX [119]. Overall, the stability of various STX variants is an unsolved analytical problem that is in need of continued study.

Significant questions and uncertainties remain with regard to the toxins associated with *P*. *parvum* at this time. Sequential extraction of cultures with various solvents in [47] identified material with cytotoxic properties, including fatty acids and fatty acid amides, but no prymnesins. However, in [63], two toxins classified as prymnesins were observed by LC–MS/MS, following multiple preparative and clean-up steps, including chlorophyll removal and post-extraction solid-phase extraction. At this point, it remains unclear which toxins are the specific causative agents for *P. parvum* toxicity, a problem exacerbated by a lack of commercially available standards from which to perform toxicity assessments.

One of the emerging needs in analytical laboratories is for methods suitable for quantifying multiple cyanotoxins in a single workflow. As cyanobacterial blooms become more common and severe, the need for rapid analysis is only going to increase and methods suitable for screening and quantifying multiple classes of toxin at once will greatly improve sample throughput. In Table 3 are listed 20 studies relying on multi-toxin methods.

One approach for extracting multiple toxins is to attempt a compromise extraction mixture that is suitable for both hydrophilic and more hydrophobic toxins, as described in [68,69,70]. In these studies, the percentage of organic solvent in the extraction mixture was reduced to ~50%, rather than the 75–90% range typical for MCs, to improve the recovery of the more hydrophilic classes of contaminants, including CYNs, ANA-a, and STXs. In [68], an acidic methanol:water (50:50) mixture was used to extract benthic algae and analysis was performed using LC–MS/MS and LC–qTOF. In this study, two STX variants were observed in field samples at 209–270 ng/g, but no detections were seen for MCs, NODs, CYN, or ANA-a. The matrix fortification of dried tissue prior to extraction found that all five classes of contaminant were recovered with 80–105% efficiency, with the caveat that the only MC congeners used were MC-LR and MC-RR, two of the most hydrophilic MCs. In [69], in contrast to a single ~50% organic extraction, sequential extractions were performed in fish and shellfish, first with 100% methanol, followed by an acidic water:acetonitrile (55:45) mixture, with the two pooled before further processing. This study was designed to be as broadly suitable as possible for screening purposes, with two chromatographic methods used, HILIC for hydrophilic toxins and reverse phase for the more lipophilic toxins. As a part of method validation, an extensive matrix spike evaluation was performed for numerous MCs, STX, and other toxins outside the scope of this review, such as domoic acid, and okadaic acid. The authors found that recoveries ranged significantly, with many within an 80–120% window but others, particularly ANA-a, showing around 200% recovery of the spiked amount. According to them, this was mostly likely the result of matrix interferences in the analysis procedure, which is not surprising given the exhaustive extraction procedure and tissue matrix, but without isotopically labeled materials, confirmation of this hypothesis is impossible. In addition, the DLs in this study were somewhat higher than typical, with DLs of 150 ng/g for MCs and 600 ng/g for the more hydrophilic toxins.

In another multi-toxin study, [71], a method to measure MCs, NOD, ANA-a, CYN, and STX in tissue using LC–MS/MS was reported. Because of differences in chemical properties, the study relied on two separate extraction procedures, one for ANA-a, CYN, and STX and the other for NOD and MCs (Table 3). Once prepared, the samples were subject to one of two LC–MS/MS methods to separate and quantify the compounds, with C18-based chromatography for NOD/MCs and HILIC for ANA/CYN/STX. As described above, the technique of isotope dilution was used in this procedure to track recovery and quantification of ANA-a, CYN, and a subset of MCs through extraction and analysis. Matrix effects for MCs were seen in fish tissue, with magnitudes ranging from −26% to −58%, which were corrected through the internal standards to −16% to 10%. Comparable matrix suppressions of −44% to −50%, corrected to 1.4 to 3.4%, were observed for ANA-a and CYN, respectively. Similar magnitudes were seen in water extraction. Extensive method validation was performed using matrix spikes to determine DLs, from which MDLs were determined to be 0.12 to 0.70 µg/kg in tissue and 4 to 80 ng/L in water. However, validation of this procedure in fish exposed to exogenous MCs in [71] was not successful in producing positive results, which the authors attributed to the short duration of exposure, but MCs were detected in environmental samples. Overall, this study illustrates the utility of the use of isotopically labeled materials in these multi-toxin methods to improve the tracking of interferences and extraction efficiencies.

Method development and optimization of techniques to extract and analyze toxins from different matrices rely on spiking the matrix with the target analytes, processing these samples through the optimized workflow, and reporting recoveries of the spiked compounds and precision as a metric of method performance. While this process may work in the case of simple matrices, such as water or algal cells, it may only partially mimic toxin-containing environmental samples from more complex matrices, such as zooplankton or tissue, as noted in [71]. For example, in spike recovery studies involving tissue, either the sample is spiked with the target toxins using a syringe needle inserted into the tissue prior to homogenization or the homogenate is spiked with the target toxins, followed by mixing prior to the processing of these samples through the optimized workflow. However, these spiked toxins may not entirely mimic the way that natural toxins are bound in tissue samples in the environment. This may lead to biases in estimated recoveries, especially if the spiked toxins are more readily amenable to the extraction process than the natural toxins, leading to incorrect quantifications of these toxins in complex matrices. While spike recovery studies are a good first step for developing and optimizing methods for toxin detection, method validation with different types of samples collected from HAB sites will provide an insight into the reliability of the methods.

**Table 3 toxins-14-00213-t003:** Selected list of publications summarizing methods for toxin detection in various matrices.

Reference	Toxins Measured	Matrix	Extraction/Sample Preparation Procedure	Analysis Method	Toxin Detection
** *Microcystins* **					
[106]	MCs	Dissolved fractions of the water table, silversides, and common carp	For tissue extraction, 75% methanol and then 75% methanol with 0.05% acetic acid was used. Samples were centrifuged, and the supernatant was blown down to dryness and resuspended in a suitable solvent for ELISA (phosphate-buffered saline), LC (methanol), and PPIA (water). The ELISA did not detect MCs within the limits of the assay, but the PPase showed that bioactive variants are present.	ELISA, LC–MS/MS, and PPIA	0.02 to 0.36 µg/L in water/sestonic, 0.16 to 0.87 µg/g in fish by ELISA, >1 µg/L PPIA
[107]	MCs	Muscle, liver, fish tissue, and lake water samples	Tissue samples were homogenized, mixed with 10 mL methanol:acidified water (90:10, *v*/*v*), and sonicated. The extracts were then centrifuged, diluted with water (not to exceed 5% methanol), and acidified. For samples with lipid >1%, an extra SPE step was included.	LC–MS/MS	349–450 ng/g in tissue
[96]	MCs	Common carp and silver carp	Tissue samples were homogenized with methanol, sonicated, and centrifuged. The supernatant was analyzed, blown down to dryness, and resuspended in a suitable solvent for analysis.	LC–PDA and ELISA	PDA: 13.8–539 µg/L in water and dry biomass. ELISA: 1.4–29 ng/g in tissue
[120]	MCs	Crab tissue	Samples were analyzed following the protocol included in the kit. This comprised tissue homogenization, extraction with methanol, sonication, and centrifugation. The supernatant was blown down prior to analysis (<5% methanol).	ELISA	Up to 1.42 µg/L in water; 65–820 µg/kg in tissue, including liver and viscera
[108]	MCs	Fish tissue	Tissue was homogenized with 3 mL of methanol, sonicated in an ultrasonic bath, and centrifuged. Supernatants were pooled and extracted with 1 mL of hexane to remove lipids. The extract was evaporated at 50 °C and reconstituted in methanol for analysis.	LC–MS/MS	<DL (1.2 ng/g) to 50.3 ng/g in tissue
[121]	MCs	Fish and crustacean tissue	Fish and crustaceans were treated with 100% methanol and then with hexane. The obtained methanolic fraction was concentrated/cleaned using SPE. The eluent was dried, redissolved in methanol, filtered (nylon filter), and analyzed by ELISA.	ELISA	0.25 to 103.3 µg/kg in tissue
[122]	MCs	Fish tissue (common carp)	Tissues were homogenized, extracted in 100% methanol, stirred overnight at room temperature, and then centrifuged. The supernatants were collected and concentrated under a N_2_ stream to 350 μL to remove the organic solvent. A 100 μL aliquot of the concentrated sample extract was diluted with 900 μL of distilled water, filtered (pore size of 0.45 μm and diameter of 4 mm), and analyzed.	ELISA	114 to 732 µg/kg in muscle, kidney, and liver
[109]	MCs	Fish tissue	Tissues were homogenized and extracted with 75% methanol. Extracts were centrifuged, the supernatant was removed, and the solids were resuspended in 75% methanol for two more extractions. The supernatant from all extractions was pooled and diluted to one-quarter strength with deionized water. The resulting solution was concentrated/cleaned with SPE (C18 column) and eluted with 5 mL of 100% methanol. The sample was then diluted to <5% methanol and analyzed.	ELISA	<7.5 to 203 ng/g in tissue
[78]	MCs	Fish tissue	Lemieux oxidation reactions were performed to convert MCs to MMPB. After termination of reactions, the samples were centrifuged at 3000 rpm for 5 min to remove tissue. Aliquots of oxidation products in the supernatant were dried and dissolved in a 5% HCl methanol solution, followed by heating and neutralization with silver carbonate. Total MC content was measured by headspace by polydimethylsiloxane/divinylbenzene (PDMS-DVB) solid-phase microextraction (SPME) GC/MS/MS analysis.	SPME–GC–MS/MS	0.018 to 0.87 µg/g by ELISA; 0.84 to 4.7 µg/g by MMPB oxidation
[51]	MCs	Rainbow trout tissue, liver, kidney	MCs were oxidized to MMPB. Total MCs were quantified using isotope dilution with d3-MMPB by LC–qTOF MS.	LC–qTOF–MS	MDL 2.18 ng/g (wet wt); tissue < MDL to 8.3 ± 6.9 ng/g; liver < MDL to 173.1 ± 97.8; kidney < MDL to 209.9 ± 42.3 ng/g
[49]	MCs	Water, fish, and mussels	A gram of water or fish was added to a centrifuge tube, spiked with MC-LR (standard addition), mixed with solvent (methanol, aqueous methanol, or aqueous acetonitrile), vortexed, centrifuged, cleaned as per [20,107,123,124], evaporated to dryness, re-constituted, and analyzed.	ELISA	0.1 µg/L (MDL)–0.2 µg/L in fish
[97]	MCs	Fish tissue	Tissue was extracted with 80% (*v*/*v*) aqueous methanol and centrifuged and the supernatant was filtered through a 0.2 μm filter. The supernatants were separated and evaporated to dryness at 40 °C. The samples were reconstituted in 300 μL of ultra-pure water for ELISA and the same amount of 100% methanol for HPLC.	ELISA and LC–PDA	0.043–1.72 mg/kg in tissue, 7.0–17.6 mg/kg in sediment, 2.9–13.5 µg/L in water
** *Nodularins* **					
[110]	NOD	Flounder, mussel, and clam tissue	Tissue was homogenized, frozen at −30 °C, and freeze-dried. Dry samples were extracted with 100% methanol and centrifuged. The supernatant was collected, concentrated at 50 °C, centrifuged, diluted 10x with Milli-Q water, filtered, and analyzed.	LC–MS/MS, MALDI–TOF–MS, LC–UV–MS/MS, and ELISA	Up to 1.490 mg/kg MCs and/or NOD in tissue by ELISA; NOD confirmed but not quantified by LC–MS
[105]	NOD	Sediments, mussels, and fish	A freeze-dried sample was ground with a mortar and pestle, extracted with 75% methanol, sonicated, and centrifuged. The supernatant was evaporated to dryness and dissolved in Milli-Q water. The sample was then vortexed, sonicated, centrifuged, and cleaned up with SPE. The cartridge was eluted with 100% methanol, dried, and re-suspended in Milli-Q water prior to analysis.	ELISA and LC–MS/MS	2.3–75 µg/kg in sediment; up to 139 µg/kg in mussels. 489 µg/kg in liver, 21 µg/kg in guts, and 21 µg/kg in flounder
[125]	NOD	Flounder and cod	Samples were extracted in water:methanol:n-butanol 75:20:5, *v*:*v*:*v*, in an ultrasonic bath for 8 h at 50–60 °C. Then, the samples were centrifuged, supernatant was evaporated to 1.2 mL, and (LC-PDA only) cleaned/concentrated with SPE and eluted with methanol. The eluent was evaporated at 50 °C to dryness, dissolved in 150 µL of methanol, filtered, and diluted with water for analysis.	LC–PDA, ELISA, and PPIA	30–70 µg/kg in liver by ELISA and PPIA, <DL of LC–PDA
** *Anatoxin-a* **					
[52]	ANA-a	Phytoplankton, stomach contents of birds, and blooms	Samples were extracted with ethanol:acetic acid (20:80) and centrifuged. The supernatant was used for assay.	ELISA	ACE inhibition equivalent to 4 µg/g ANA-a in extracts; 0.1 to 0.9 µg/g MCs by ELISA
[53]	ANA-a	Water samples/ slurry	Immunoaffinity beads were employed for the extraction of ANA-a from water. Sample pH was adjusted to 10, then a magnetic immunosorbent was added to the sample and mixed for 10 min. The magnetic particles were separated rapidly from the solution by an external magnet, and the water sample was gently removed. Then, ANA-a was completely eluted with 2-propanol. The solution was separated from the magnetic particles by an external magnet and directly analyzed by IMS.	IMS	0.02 to 5 µg/L linear range by IMS
[54]	ANA-a	Water	SPE cartridges were conditioned with 2-propanol, followed by HPLC-grade water. The samples were applied to the cartridges dried under vacuum and the analyte eluted with methanol containing 0.1% v/v trifluoroacetic acid. The extracts were blown down to dryness at 40 °C, re-dissolved in 5 % v/v aqueous acetonitrile containing 0.1% v/v trifluoroacetic acid, and analyzed.	LC–UV	DL of 25 ng/L
** *Saxitoxin* **					
[55]	STX	Shellfish tissue	Samples were extracted using 0.1 M HCl with ultrasonication, cleaned/concentrated with SPE (C18 cartridge), and analyzed.	LC–qTOF MS	0.1–1.6 µg/kg recovery from spiked tissues
[58]	STX	Sheep intestine and blood	Samples were sonicated with 0.1 M acetic acid and incubated for 2 h at 4 °C. Clean-up was performed with a C18 cartridge; 1 mL of 0.05 M acetic acid was used to elute the toxin fraction.	LC with spectrofluorometric detector	STX detected in intestine but not in blood; exact concentration not reported
[56]	STX	Seabird tissues, forage fish, and invertebrates	Seabird tissues and whole forage fish and invertebrates were extracted for STX analysis using the procedure of [117]. Tissue was homogenized, extracted in 3 mL of 1 % acetic acid, vortexed, boiled, allowed to cool to room temperature, vortexed again, and centrifuged. The remaining supernatant was poured into a vessel; the procedure was repeated. The combined supernatant was filtered and diluted in Milli-Q water for analysis.	ELISA and LC–FLD	0.14–1.08 µg/kg in liver by ELISA; no detection using LC–FLD with DL of 1 µg/kg
[59]	STX	Bivalves	A tissue homogenate (1.0 g) was mixed with 5.0 mL of phosphate buffer solution in a 50 mL plastic centrifugal tube and then placed in a boiling water bath for 5 min, cooled, extracted in ultrasonic water bath at room temperature, and centrifuged. The supernatant was collected, and the residue extracted once more. The supernatant was combined and filtered by microfiber filters. The filtrate was cleaned using immunoaffinity column (IAC). The eluent from the IAC was blown dry with N_2_ at 55 °C, redissolved with 1 mL of water, and filtered by a 0.22 μm membrane before determination by LC–MS/MS.	LC–MS/MS	DL 0.1 μg/kg
[57]	STX	Abalone	About 2 g of abalone tissue (epipodium, viscera, or foot muscle) was mixed with 18 mL of 1% acetic acid (v/v). The mixture was vortexed, boiled, cooled, vortexed again, and centrifuged. This was followed by the addition of 5 µL of ammonium hydroxide before SPE clean-up. STX was eluted using 2 mL of acetonitrile:water:acetic acid (20:80:1, v/v/v) and diluted with acetonitrile before analysis.	LC–MS/MS	High detection in muscle/epipodium (up to 1.085 mg/kg) exposed to STX producing cultures
** *BMAA* **					
[60]	BMAA	Cyanobacterial samples	The lyophilized sample was hydrolyzed using 6 N HCl liquid hydrolysis for 20 h at 105 °C in the absence of oxygen. After hydrolysis, samples for derivatized analysis were dissolved in 500 µL of hot 20 mM HCl and subsequently diluted 10 times in HCl to obtain a protein concentration below 0.1 g/L. Hydrolyzed samples for underivatized LC–MS/MS analysis were dissolved in 1 mL of 65% acetonitrile, 35% Millipore water, and 0.1% formic acid (*v*:*v*:*v*).	LC–FLD and LC–MS/MS	ND by LC–MS/MS, false positives by HPLC–FLD
[61]	BMAA	Water samples and tissue samples (crustacean, mollusk, and fish)	Centrifuged, homogenized tissue was suspended in trichloroacetic acid and washed with chloroform for the removal of residual lipids. Samples (5 mL) and standards were derivatized with 6-aminoquinolyl-N-hydrosuccinimidyl carbamate (AQC), and BMAA was separated from the protein amino acids by reverse-phase elution (Waters Nova-Pak C18 column). Identification of a BMAA peak detected by reverse-phase HPLC was verified by LC–MS/MS using product ion mode in a triple quadrupole system.	LC–FLD	<DL to 7 mg/g by FLD; confirmed by LC–MS/MS
[62]	BMAA	Freshwater surface samples, mollusks, crustaceans, and fishes	The lyophilized sample was extracted with 2 mL of 0.1 M trichloroacetic acid by sonication in an ice bath. The extract was centrifuged, and the supernatant was N_2_ dried for the collection of free BMAA. The precipitated protein pellets were subsequently hydrolyzed in 6 M HCl and filtered. The hydrolysate was then N_2_ dried for protein-associated BMAA collection. The free and protein-associated BMAA fractions were reconstituted in 20 mM HCl. Samples were derivatized by adding 60 μL of borate buffer and 20 μL of AQC. The mixture was incubated in a water bath at 55 °C for derivatization and was prepared for LC analysis.	LC–MS/MS	0.45–6.05 µg/g dry weight
** *Cylindrospermopsin* **					
[111]	CYN	Fish tissue and liver	Tissue and liver were homogenized in 10 mL of 100% methanol, sonicated, and centrifuged. The supernatant was decanted and filtered. The extraction was repeated on the pellet, and the two extracts were collected together and then dried by rotavapor at 40 °C; the residue was re-suspended in 2 mL of distilled water and analyzed.	LC–MS/MS and ELISA	2.6 to 126 µg/L in water; up to 2.7 ng/g in fish tissue
[112]	CYN	Crayfish tissue	Freeze-dried samples of cyanobacteria and tissue were taken up in distilled water with sonication, filtered, and diluted to a concentration within the linear range of the method. Water samples were filtered and diluted when necessary.	LC–MS/MS	589 µg/L in water; up to 4.3 and 0.9 µg/g in liver and muscle tissue, respectively
[113]	CYN	Mussel	Lyophilized tissues and samples for the analysis of intra- and extracellular CYN were extracted in 100% methanol with ultrasonication on ice. Tissue and cell debris was removed by centrifugation and the supernatant dried at 50 °C under N_2_ and re-suspended in Milli-Q water. The samples were centrifuged again to remove insoluble materials and analyzed.	LC–UV	Up to 2.52 µg/g in tissue
** *Prymnesin(s)* **					
[126]	*P. parvum* strains	Water and algal cells	Samples were placed in 15 °C and incubated at an irradiance of 5–7 mmol photons m^2^ s^1^ for 2 h. After 2 h, the in vivo fluorescence of the samples was measured on a Turner Design Trilogy1 Laboratory Fluorometer.	Relative fluorescence	Toxin extracts highly unstable when extracellular; storage at −80 °C with no headspace indicated
[47]	*P. parvum* strains	Water and algal cells	Water and cultured and field-collected algal cell mass was lyophilized. An elutropic extraction scheme using solvents with increasing polarity (dichloromethane, ethyl acetate, methanol, and water) was used to fractionate toxic compounds in samples by polarity. Individual compounds were obtained via semi-preparative HPLC–MS purification. Isolated compounds were then structurally characterized by MS/MS and NMR.	LCMS/MS, LC–HRMS, and NMR	Structural identification of potentially toxic compounds in extracts
[92]	*P. parvum* strains	Water and algal cells	Liquid–liquid partitioning of the whole cultures (medium plus cells) using ethyl acetate was performed. The ethyl acetate layers from partitioning against 50 L of *P. parvum* cultures were combined, and the organic extract was subjected to gradient MPLC.	GC–MS and NMR	Additional structural characterization of potential toxins
[127]	*P. parvum* strains	Water and algal cells	Samples were preserved with acid Lugol’s solution, and cells were counted using a particle counter.	Cell density	Characterization of parameters influencing toxicity of *P. parvum* cells
[46]	*P. parvum* strains	Water and algal cells	The biomass pellets were thawed and extracted twice with cold acetone for removing, among other, chlorophylls. After vortexing and centrifugation, the supernatants were collected (acetone). After chlorophyll extraction, the biomass was extracted twice with methanol and sonicated. Both extracts (acetone and methanol) were concentrated to dryness under N_2_ at 35 °C, reconstituted in 1 mL methanol, and analyzed.	LC–DAD–HRMS	Prymnesins characterized and identified
[63]	*P. parvum* strains	Water	The samples were extracted with cold acetone, methanol, and isopropanol. This was followed by pooling of samples, SPE, and evaporating the eluent to dryness. The dried methanol:isopropanol fraction was resuspended in water. An equal volume of ethyl acetate was added to the sample and placed on the vortex mixer, followed by centrifugation. The aqueous portion was recovered and defatted with ethyl acetate three more times. After the last phase of partitioning, the aqueous layer was transferred back to the methanol:isopropanol vial and evaporated to dryness. This was followed by SPE and analysis.	Thin-layer chromatography (TLC) and LC–HRMS	Prymnesins characterized and identified
** *Concurrent Analysis of Multiple Cyanotoxins* **					
[67]	MC NOD ANA CYN STX	Fish from aquaculture	MCs, NOD, ANA, and CYN: Toxins were extracted twice with a water:methanol mixture (50:50, *v*/*v*), followed by 10 min sonication in an ultrasonic bath, and then treated for 2 min in an ultrasonic homogenizer. The extracts were centrifuged (14,000 rpm), and the supernatant was and analyzed via LC–ESI–MS. STX: By adding 1 mL of acetic acid (0.03 N), 50 mg of lyophilized samples was extracted, sonicated in an ice bath, and centrifuged. The supernatant was then filtered, and the extract was analyzed via LC–FLD.	LC–MS/MS and LC–FLD	No detection of MCs, NODs, ANA, or CYN by LC–MS/MS; up to 350 ng/g STXs by LC–FLD
[68]	STX ANA NOD MCs CYN	Benthic *Lyngbya wollei* algae samples	Dry algae were mixed with 1 mL of methanol and water (1:1) with 0.1 M acetic acid. Samples was vortexed, sonicated, and centrifuged. The supernatant was collected and filtered through a 0.45 µm PTFE filter. This procedure was repeated three times in total. All aliquots were combined, evaporated to dryness under N_2_, and resuspended in 0.5 mL of acetonitrile:water (9:1) with 5 mM ammonium acetate and 3.6 mM formic acid (pH 3.5) for analysis.	HILIC and RPLC–MS/MS, LC–QqQMS, and LC–QqTOFMS Ɨ	209–279 µg/g of two STX analogs (LWTX-1 and LWTX-6) in algae; no other cyanotoxins detected
[128]	MC ANA	Lake water samples and freeze-dried bloom material	Lyophilized cells (about 100 mg) were extracted three times with 10 mL of 0.05 M acetic acid for 30 min while stirring. The extract was centrifuged, and the supernatant was adjusted to pH 10 with 7% ammonium hydroxide. This pH 10 extract was directly applied to 0.2 g of a reversed-phase ODS disposable extraction column.	LC–PDA	20–1500 µg/g of various MCs; up to 1444 µg/g ANA
[64]	MCs NOD	Cyanobacterial bloom material obtained from freshwater lakes	Samples were lyophilized, and extracts were prepared using 70% (v/v) methanol and centrifuged. The resulting supernatants were analyzed.	PPIA and LC–PDA	DLs of as low as to 1 µg/L in drinking water
[65]	MCs NOD	Otter tissue, digesta, and water	Tissue samples were first homogenized, mixed with methanol:water (90:10), sonicated, and analyzed. Sample preparation and analysis followed protocols from previously published studies [107].	LC–MS/MS	1.36–348 µg/kg in otter liver; up to 1324 µg/kg in clams, mussels, and oysters
[66]	MCs NOD	Water, algal cells, algal supplement tablets, and mussels	Water samples were analyzed directly by LC without any extraction steps. Algal samples were centrifuged to isolate cells. The cells, tablets, and mussel tissue were extracted using a variety of solvents in different proportions (aqueous methanol, isopropyl alcohol, and 1% acetic acid). It was found that 80% aqueous methanol enabled the optimum extraction of toxins. Samples were extracted by vortex mixing.	LC–MS/MS	Limit of detection ranging from 0.01 and 0.19 ng/mL for water, 0.4 and 3.6 pg/mL for algal cells, 0.12 to 1.18 μg/kg for algal supplement tablet powders, and 0.01 and 0.21 μg/kg for mussels
[129]	MCs NOD	Bottlenose dolphin liver	Samples were oxidized to convert MCs/NODs to MMPB. Samples were cleaned with SPE and 12 cc Novum simplified liquid extraction (SLE) tubes and analyzed with LC. Individual variants were extracted with 75% methanol in 0.1 M acetic acid, followed by a butanol rinse. Supernatants were blown to dryness using N_2_ at 60 °C, reconstituted in deionized water, clarified using SPE, eluted with acetonitrile, blown to dryness (60 °C, N_2_), reconstituted (1 mL of 5% methanol), filtered (0.2 μm polyvinylidene fluoride), and analyzed using LC. The final extract was also diluted 10-fold for ELISA analysis.	LC–MS/MS and ELISA	MDL 1.3 ng/g for the MMPB method and 1.6–11.5 ng/g for the variants
[69]	MCs STXs CYNs Others	Fish, shellfish tissue, and food supplements	A gram of tissue homogenate was extracted with 4 mL of methanol with a vortex mixer and centrifuged and the supernatant decanted. To the pellet, 5 mL of water/acetonitrile/ammonium formate/formic acid (55:45 v/v, 2 mM, 0.5 mM) was added and extracted with a pulse mixer and centrifuged. The supernatant was combined with the previously obtained methanol extract. The tube was filled with 10 mL of acetonitrile. The aliquot of the extract was filtered with a 0.2 µm filter and used for analysis with LC–HRMS. The supplements followed a similar procedure with an additional clean-up step using a Strata-X polymeric reversed-phase cartridge.	LC–HRMS	DLs of 150 ng/g for MCs and 600 ng/g for the more hydrophilic toxins; 80–200% recoveries
[130]	MCs NOD ANA CYN	Water, fish tissue, and liver	Water: Of the sample, 150 mL was filtered using a glass fiber filter, adjusted to pH 11, and cleaned with SPE. Eluents were evaporated to dryness under N_2_, reconstituted with 150 µL of 5% (v/v) methanol, and then analyzed with LC. PIPPA and ELISA analysis was performed using commercial kits as per manufacturer-provided guidelines. Tissue/liver: With 5 mL of 80% methanol containing 0.5% formic acid, 0.2 g of lyophilized powdered flesh or 0.25 g of liver was extracted by stirring for 15 min, followed by ultrasonication for 30 min. The mixture was centrifuged, and the supernatant was washed three times with 1 mL of hexane. The extract was cleaned by SPE. The eluents were dried in a water bath at 40 °C under N_2_, reconstituted with 200 µL of 5% methanol, and analyzed with LC.	LC–MS/MS, PPIA, and ELISA	25.8–429.3 μg/L MCs in water; no detection in tissue
[114]	MCs ANA-a CYN	Fish tissue	Tissue (500 mg wet weight) was amended with 150 μL of a mixture of isotope-labeled internal standards. After a 1 h equilibration time, 4 mL of methanol was added, vortexed, ultrasonicated, and centrifuged. The supernatant was removed, and the tissue was re-extracted twice as previously described. The combined supernatants were concentrated to 4 mL (N_2_, 40 °C). The samples were then frozen and centrifuged (defatting step), and the supernatants were evaporated to dryness (N_2_, 40 °C), reconstituted in 2 mL water, vortexed, ultrasonicated, filtered (0.2 μm), and analyzed.	Online SPE–LC–MS/MS	MDL of 0.1 to 10 μg/kg; 0.16–7.8 μg/kg of MCs and 46 μg/kg of CYN detected in field samples
[131]	MCs NOD CYN STX	Carp, otter, dalmatian pelican tissue, and liver and stomach contents	Samples were freeze-dried, ground using a pestle and mortar, and extracted three times at 60 °C in 0.5 mL of 75:25 methanol:water (*v*:*v*). Extracts were dried in SpeedVac and reconstituted in 600 μL of methanol. The reconstituted samples were transferred to 2 mL of Eppendorf vials with a cellulose acetate filter and centrifuged for 5 min. Filtrates were transferred to amber glass vials for MC analysis.	LC–MS/MS	0.8–1.9 μg/g of MCs in carp liver; 0.7 μg/g MCs in otter liver; 0.4–1.5 μg/g MCs in pelican liver, tissue, and stomach sample
[71]	MCs ANA STX NOD CYN	Fish tissue	Homogenized whole fish, 2 g, was lyophilized in a freeze dryer for 72 h. ANA, CYN, and SAX were extracted with 10 mL of 25:75 (*v*:*v*) acetonitrile:water added to each vial. MCs and NOD were extracted using 10 mL of 75:25 (*v*:*v*) acetonitrile:aqueous 0.1% formic acid added to each vial. Samples were sonicated and centrifuged. The supernatant was collected, syringe-filtered, blown down under N_2_, and re-suspended in 20 mL of water. SPE was performed for clean-up (ANA, CLD, and SAX were extracted on a Supelclean ENVI-carb. MCs and NOD were extracted using an Oasis HLB). Analysis was performed using LC–MS/MS.	LC–MS/MS	Non-detection in fish exposure study method; MDLs from 80 to 960 ng/L in water and 0.12 to 0.70 µg/kg in tissue
[132]	STX ANA	Phytoplankton samples	Freeze-dried material (10 mg) and 2 mL of 0.03 N acetic acid were mixed, frozen and thawed three times, sonicated, and centrifuged. The supernatant was filtered and stored at −20 °C until analysis.	LC–FLD	5.9–224.1 ng/g STX equivalents
[20]	MCs NOD	Fish	Homogenized fish tissue was weighed, extracted with a 3:1 methanol:water solution with 1% formic acid, vortexed, centrifuged, extracted with hexane clean-up to reduce lipid content, centrifuged, and analyzed.	LC–MS/MS and ELISA	10 ng/g DLs in tissue
[133]	STX NeoSTX GTX(1,2,3,4,5)	*Hoplias malabaricus* (wolf fish)	The samples were homogenized in HCl (0.1 N) and centrifuged at 10,000 ×g at 19 °C for 10 min. The supernatants were filtered with cellulose filters and analyzed.	LC–FLD	No detections of STXs in the tissue after exposure
[116]	BMAA DABA ANA-a	Water fish aquatic plants	Samples were mixed with 1 mL of 0.1 N TCA by vortexing for 1 min and washed with 100% purified water, 50% methanol in water, and 100% methanol. The mixture was then vortexed and centrifuged to separate solids from the aqueous extract. The extract containing unbound or “free” amino acids, including BMAA and DABA (2,4-diaminobutyric acid), were transferred to a microcentrifuge filter tube for removal of suspended proteins and centrifuged before analysis.	LC–FLD and LC–MS/MS	BMAA between 8 and 59 ng/g in tissue; no ANA-a detections reported; BMAA, DABA, and ANA-a detected in plants
[134]	BMAA DABA ANA-a	Lake water, fish, and aquatic plants	Freeze-dried samples were ground into a fine powder and extracted with 0.1 N TCA. The mixture was sonicated for 30 s, refrigerated for 16 h, and centrifuged and the supernatants retained. The process was repeated once more. The supernatants were combined, filtered, and analyzed with HPLC–FLD for preliminary analysis of all extracts; confirmation was performed using LC–MS/MS.	LC–FLD and LC–MS/MS	0.8–3.2 µg/L DL for LC–MS/MS; 5–7 µg/L for FLD
[135]	STX NOD MCs	Fish tissue	Freeze-dried muscle tissue was extracted with methanol, sonicated, and centrifuged and the supernatants retained. For lipid removal, hexane was added to the supernatants and then discarded after phase separation. Samples were evaporated and 10% methanol was added, followed by sonication and passage of the material through reversed-phase cartridges (OASIS HLB Cartridge 200 mg, Waters). Cartridges were eluted with 100% methanol, followed by evaporation and dissolving the residues in 75% aqueous methanol. After vortexing, the samples were filtered and centrifuged. The supernatants were then diluted 10-fold with 75% methanol for analysis.	LC–MS/MS	Detection of STXs, NODs, and MCs in water, while only MC-RR detection in tissue
[77]	MCs ANA-a	Fish tissue	Two types of ELISA kits were used for samples: Envirologix™ anti MC-LR and Abraxis LLC anti-adda. The samples were extracted with methanol:water followed by SPE clean-up, similar to [107]. LC–MS/MS was used for confirmation.	ELISA and LC–MS/MS	2.2 to 132 µg/kg by anti-adda ELISA; 0.2–2.4 by anti-MC-LR ELISA; 2.5–14 µg/L MC-LA by LC–MS/MS; potential false positive detection by adda-ELISA
[70]	MCs ANA	Cyanobacterial biomass and fish tissue	Cyanobacterial biomass and fish tissues were prepared in acidified (0.002 M HCl) 50% methanol. Both biomass and tissues were homogenized, ultrasonicated (3 times), and treated with n-hexane to remove lipids (hexane layers were discarded). The obtained methanol extracts were analyzed.	LC–PDA	Up to 18.4 µg/g ANA and up to 4.4 µg/g MCs in liver tissue

Ɨ Hydrophilic interaction liquid chromatography (HILIC), reverse-phased liquid chromatography (RPLC) coupled to triple quadrupole mass spectrometry (LC–QqQMS), and quadrupole–time of flight mass spectrometry (LC–QqTOFMS).

## 6. Conclusions and Recommendations

This review documents the methodologies currently used to measure cyanotoxins and prymnesins in complex matrices and to assess the advantages and limitations of the various techniques summarized in this paper.

Due to the diversity in chemical structures and properties among cyanotoxins, it is a significant challenge to develop and validate procedures for consistent and high-yield extraction from various matrices, including tissues. In individual toxin methods, optimum conditions can typically be achieved through validation experiments, such as the consensus for extractions using 75–90% methanol:water for MCs/NODs consistently seen in the literature, although variations in matrix constituents such as lipid or protein content could potentially result in varied outcomes. The use of matrix spikes, where known amounts of toxins are fortified to sample matrices, can help further quantify the various matrix effects and interferences in extraction and analysis of cyanotoxins and add confidence to an analytical workflow. This is particularly important in cases where ambient detection of toxins was negative, as in studies such as [55], where only spiked samples showed detectable levels of toxins to allow for evaluation of recovery efficiencies.

As the breadth of cyanotoxins known to be present in the environment continues to increase, it is clear that methods for quantifying single classes of toxin may eventually overwhelm analytical facilities. A single method capable of extracting and detecting many or all cyanotoxins would be an ideal goal for the future. Several publications describe such approaches using mass spectrometric detectors for a subset of contaminants of interest, but as expected from a group of compounds with diverse chemistry, there are significant limitations in recoveries during sample processing, chromatographic performance, and sensitivity [104,105,106,107,108,109,110,111,112,113]. For example, in [69], MCs, NOD, ANA-a, CYN, and STXs were all analyzed using LC–MS/MS following extraction, with sequential extraction first with methanol, then a mixture of acetonitrile:water (45:55), which were pooled for analysis. Method recoveries in this study varied between 80 and 200% and the reported detection limits were high, most likely due to matrix interferences. In other cases, multiple complementary workflows for hydrophilic and hydrophobic toxins were developed, as in [71], and showed improved performance.

To control for variation in extraction efficiencies, one approach is to add isotopically labeled internal standards to the sample prior to extraction, allowing for quantification of recovery percentages through the extraction and analysis procedures. This can help compensate for recovery bias and matrix suppression/enhancement and is a recommended best practice but is limited in scope to those compounds for which isotopically labeled materials are readily available and when analytical methods can differentiate the native and labeled material (e.g., mass spectrometric methods). Recently, labeled materials have become available for a broader set of cyanotoxins, including MCs, STX, ANA-a, and CYN, which has made this approach feasible. Examples of this as applied to toxin measurement can be found in reference [71], where recoveries were assessed and corrected for both extraction efficiency and matrix effects using labeled toxin analogs.

Analysis with LC–MS/MS has become an essential tool for cyanotoxin detection and it can potentially be used for the concurrent analysis of multiple classes of toxins due to its rapid scan rate and ability to cycle polarity from positive to negative ion modes. LC–MS/MS is best employed for targeted screening for toxins, particularly where both native and isotopically labeled standards are available. In contrast, LC coupled with HRMS or TOF is better suited to detect unknown toxins or those for which standards are not available, but a more complete discussion is beyond the scope of this article [82].

An alternative to LC–MS/MS methods could be an ELISA microplate test strip that contains antibodies for multiple toxins. This would require design in a way that similar incubation periods for the binding step could be achieved, as the typical ELISA workflow has specific duration of each step of binding and rinsing of the plate. If this was achievable, it would allow simultaneous testing for multiple toxins without requiring expensive and bulky experimental apparatus, such as mass spectrometers. Because ELISA methods tend to be semi-quantitative for reasons described above, it is a best practice that detections be confirmed by LC–MS/MS or some other more specific technique where possible. One potential alternative in the future could be electrochemical biosensors that contain a biological recognition element that specifically reacts with the target of interest; these are in active development for use with water samples. The suitability of these biosensors to fish tissue and other matrices still needs to be determined.

Another complicating factor for assessment of toxins in tissue matrices is common to any analytical workflow, namely, the amenability of these bound toxins to be extracted into a solution for measurement. Literature results suggest a significant fraction of cyanotoxins in tissue samples could be bound to organs or otherwise unavailable through the extraction procedure [106]. Ref. [78] showed significant differences in MC measurements in tissue using ELISA and GC–MS/MS following Lemieux oxidation, which they attributed to the oxidation technique freeing bound analytes for measurement. Ref. [106] and references therein suggest that the total concentration could potentially be an order of magnitude higher. The level of underestimation and impact on risk assessments and health outcomes should be investigated in future studies.

An additional challenge is that many emerging cyanotoxins do not have commercially available standards, and, in other cases, toxin standards are prepared from minute amounts of natural sources or unidentified sources of unknown purity, making it difficult to accurately quantify toxin concentrations. In this review, prymnesins and other potential toxins produced by *P. parvum* are one such class of toxins, which at present are only available through laborious culturing, extraction, and isolation, as described above, and no commercial sources exist. Even in cases where materials can be procured, the use of certified reference standards (CRMs) with exact concentrations is recommended to improve confidence in the absolute concentrations; however, these are not available for many contaminants, and even for those with CRMs available, such as MCs, only a subset of variants may be covered. It is recommended that laboratories monitor their cyanotoxin standards over time for variations in purity/concentrations and where possible obtain standards from multiple vendors if certified materials are not available.

There is a great deal of variability in the analytical procedures presently being used to prepare, extract, and analyze for cyanotoxins associated with harmful algal blooms in diverse sample matrices. In this review, the most common procedures were highlighted, and best practices were identified. It is clear that there is a compelling need for more standardized, reliable, and affordable screening methods compatible with tissue and similar matrices exposed to cyanotoxins, particularly as new toxins are continually being identified in the environment. For LC–MS/MS methods, the best approach is to ensure the use of extensive quality control procedures, including evaluating matrix interferences, though matrix spikes where possible and using labeled surrogate and internal standards to monitor method performance across both the extraction and analysis phases (e.g., [71]). For the commonly used immunological methods, because they are incompatible with the use of labeled standards, researchers should instead ensure they perform similar studies of matrix performance to determine cross-reactivity parameters and potential interferences as part of their method validation procedures.

## Figures and Tables

**Figure 1 toxins-14-00213-f001:**
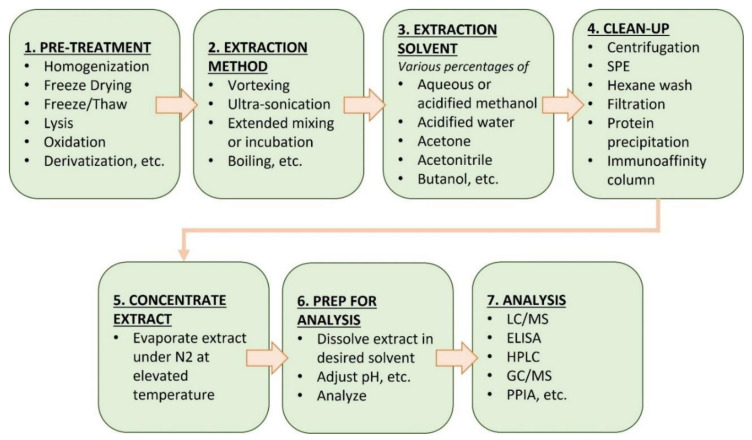
Common sample extraction and preparation procedures for measuring cyanotoxins in tissue.

**Table 1 toxins-14-00213-t001:** Characteristics of the most commonly studied cyanotoxins and prymnesins.

Common Name	Chemical Group	Effect/Target in Mammals
Microcystins	Heptapeptide	Cytotoxicity; genotoxic effects in liver
Nodularins	Pentapeptide	Cytotoxicity; liver
Anatoxin-a	Bicyclic amine alkaloid	Neurotoxicity
Saxitoxins	Tricyclic perhydropurine alkaloids	Neurotoxicity
β-N-methylamino-L-alanine	Amino acid	Neurotoxicity
Cylindrospermopsins	Polycyclic uracil with guanidine and sulfate group	Multitarget alkaloids
Prymnesins	Polyether polycyclic core with several conjugate double and triple bonds	Hemolytic activity and ichthyotoxicity

**Table 2 toxins-14-00213-t002:** Analytical methods available for the detection of cyanotoxins and prymnesins (primarily used for water but adapted for use with fish tissue and other matrices).

Common Name	Commonly Used Analytical Detection Techniques
Microcystins	Immunoassay, LC–PDA, LC–MS *, GC–MS, PPIA
Nodularins	Immunoassay, LC–PDA, LC–MS *, PPIA
Anatoxin-a	Immunoassay, LC–UV, LC–MS *, IMS
Saxitoxins	Immunoassay, LC–FLD, LC–MS *
BMAA	Immunoassay, LC–FLD, LC–MS *
Cylindrospermopsins	Immunoassay, LC–UV, LC–MS *
Prymnesins	MS–MS, HRMS, qTOF, NMR

* Includes LC–MS, LC–MS/MS, and high-resolution mass spectrometric technologies.

## Data Availability

Not applicable.

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
