# Peer review of "Determination of Cyanotoxins and Prymnesins in Water, Fish Tissue, and Other Matrices: A Review"

_toxins, 2022, doi:10.3390/toxins14030213_

Round 1

Reviewer 1 Report

My comments are in the attached file.

Reviewer 2 Report

The aim of this paper was to review all of the different methods available for cyanotoxin detection in multiple different matrices. It clearly outlines the need for a standardized extraction methods which is a pitfall for cyanotoxin research outside of detection in water. This paper does a good job briefly going over the different types of methods that are used for extraction and detection, it lacks in depth and detail of these methods throughout the paper. 

This review would be stronger if it also summarized what extraction techniques seem to give the best recoveries. Furthermore, try to answer questions like what are advantages and disadvantages to using specific extraction solvents. More detail on methods that are able to analyze more than one group of cyanotoxins, and what compromises to the analysis were needed in order to make it a successful method are also needed.

Throughout the document there are multiple grammar issues as well as a lack of consistency when using abbreviations in comparison to the words that you are using the abbreviations for, and proper spacing with units. Also, watch where you are defining the abbreviations. There are multiple instances in which they are defined more than once or not at all.

Areas to address.

1) Table 1: take out molecular formula and molecular weight. All of the cyanotoxin types have multiple congeners. If you are going to keep them, then I would suggest using a range. But it would be better without those two columns.

2) Section 2: Give more in depth representation for each cyanotoxin. Make a list of key points that you want to explain, and check it off for each cyanotoxin. There were gaps in their representation.

3) Lines 117-119: Italicize species names.

4) Line 148: Why is there an equal sign before Prymnesiophyta? 

5) Line 350-353: This sentence needs to be restructured because it is coming off miss leading. LC-MS/MS is more sensitive than LC-PDA, LC-UV, and LC-FLD. Furthermore, LC-MS/MS is more advantageous because you can look at ion ratios to determine if the cyanotoxin that you are observing is the same as the standard or different. This is important so that there are no miss identifications occurring in the analysis. 

6) In Section 4.4 you have not defined protein phosphatases, though an abbreviation is being used.

7) Table 3: This table needs to be reformatted. 1) Grammar and spelling checks, as well as checking for sentence integrity. This is crucial for the extraction/sample preparation procedure column.  2) Chemical abbreviations used throughout but so are the words. Make it consistent. Define as a footnote. 3) Example: µ vs u. Make sure they are all µ. 4) Spacing of the units and consistency of the units. There are a lot of units without spacing from the number amount. For example, mL and ml. 5) Consider making the table larger and landscape view. Spacing between columns is needed. 

8) Table 3: Consider adding another couple columns. One column that contains analyte recoveries. It is good that there have been multiple extraction methods for different matrices summarized, but it would also be good to see if they are extracting well. The other column that would be helpful would be how they prepared their standard or used a surrogate and internal standard for the analysis. Or how did they matrix match their curve to fit the sample analysis. These are just ideas to make it stronger.

Reviewer 3 Report

This is a generally interesting review on the methodologies used to measure the most commonly found freshwater algal toxins in various matrices, discussing also their advantages and limitations. The manuscript is comprehensive and summarizing ample information on the topic, however its structure, as well as certain details, need some revisiting. For example, some more emphasis should be put on discussing the details of methods destined for matrices other than water. The specific points needing improvement in order to make the manuscript more robust and reader friendly are described below.

General remarks:
A. The article should emphasize more on the methods relevant to matrices other than water (complex matrices), which is a more under-reviewed topic, and provide more details and discussion on this issue. The text with regard to methods in complex matrices is very brief – numerous details are placed in Table 3, but not discussed further. In multitoxin methods, only two methods are discussed in detail (refs [15] and [77]) where numerous others are indicated in the relevant section of Table 3 (e.g. [39, 48, 62, 76, 104-113]). Is there a reason for this preference of the authors? The choice of these methods as representative ones should be justified.
B. Throughout the text: there are numerous cases where abbreviations are used and they are not explained at their first instance in the manuscript, but are explained much later (e.g. page 2, line 59 – SPE; page 2, line 61 – ESI, page 4, Table 2 – qTOF, etc.). Also, vice-versa, some abbreviations explained, are re-abbreviated later in the manuscript (e.g. page 2, line 49 and page 6, lines 209-210). Such discrepancies should be corrected in the whole manuscript.
C. Please check that all instances of genus and species names are italicized, there are multiple instances in the text where this is not the case (e.g. page 3, lines 117-119 and many other cases).
D. Please correct the referencing pattern according to the journal instructions, e.g. page 1, line 36: [1], [2], [3], [4], [5], [6], [7], [8], [9], [10] should be just [1-10]. There are multiple instances of this appearance within the manuscript. Furthermore, in page 19, lines 461-464, references are provided using names and not numbered. Also, the reference list should be formatted accordingly.

Specific remarks:
Title
- The title is not representative of the manuscript content. It should be clear that only methods relevant to cyanobacterial/freshwater toxins are reviewed, not algal toxins in general.

1. Introduction
- Page 2, lines 53-55: LC-UV and HPLC-PDA methods are more or less similar, they should be at least grouped together or appear closely.
- Page 2, lines 59-63: These methods (544,545,546) should be individually referenced and included in the reference list.
- Page 2, lines 63-65: Similarly, it is better to refer to the particular method number (EN 22104) and include in the reference list, this is a published method [ISO 22104:2021(en) Water quality — Determination of microcystins — Method using liquid chromatography and tandem mass spectrometry (LC-MS/MS)].

2. Cyanotoxins
- Page 3, Table 1: It should be made clear that in the “common names” column, these are actually toxin groups and not particular toxins. Each group has very many analogues. It should also be clarified that the molecular formula and MW provided refers to only one (probably the main) representative of the toxin group and it should be indicated which exactly is this (e.g. in saxitoxins (PSTs) I assume the authors are referring to saxitoxin itself and not in any of its analogues – this should be evident by the column names/details provided). Alternatively, the table should be expanded to contain more of the known analogues (this could be placed in a supplementary section).

3. Current Detection Methods for Cyanobacterial Toxins
- Page 4, Table 2: Please explain any first appearing abbreviations.

4. Sample Preparation and Analytical Methods for Algal Toxins Detection 
- Page 6, line 213: Correct “limits” to “limit”.
- Page 6, line 252: Correct “amended” to “adjusted”.
- Page 7, line 278: The authors should make sure they distinguish between LC-MS and LC-MS/MS methods. The terms cannot be used interchangeably. Could be better to refer to LC methods involving mass spectrometry detection.
- Page 8, lines 350-353: “Because … environmental laboratories”: please rephrase, the syntax is confusing.
- Page 9, line 406: What does “MDLs” stand for? It is not explained anywhere in the manuscript.

5. Conclusions and Recommendations
- Page 19, line 432: Similar to the above, the terms LC-MS and LC-MS/MS cannot be used interchangeably. Please be specific, as the methods referenced (104-113) contain both LC-MS and LC-MS/MS methods.

Round 2

Reviewer 1 Report

The authors have reviewed the manuscript accordingly. It can be published successfully.

Author Response

Thank you for the comments and suggestions. You've helped significantly improve this manuscript. 

Reviewer 2 Report

Overall, this reads much better, has better organization, and detail. Once completed, it will be a valuable paper. Thank you for addressing all of the fixes with much detail. Below are the minor edits that need to be made or addressed.

Minor edits

1) Line 28: U.S. The period was missing in the first and second draft.

2) Table one: space needed in amino acid

3) Figure 1: capitalize Immunoaffinity

4) Line 544, refer to the lead authors names, or find another way to describe the studies. The way it is done is not proper. Please do the same for lines 610 and 613.

5) Throughout the document there are areas in which microcystin is still written out and not abbreviated.

6) Cylindrospermopsin showing both CYL and CYN abbreviations. I think CYL is the most common.

7) In the discussion, section 5.0, Prymnesins are not discussed. Why is that?

Author Response

Review Comments:

1: The authors have reviewed the manuscript accordingly. It can be published successfully.

2. Overall, this reads much better, has better organization, and detail. Once completed, it will be a valuable paper. Thank you for addressing all of the fixes with much detail. Below are the minor edits that need to be made or addressed.

Minor edits

1) Line 28: U.S. The period was missing in the first and second draft.

Corrected

2) Table one: space needed in amino acid

Corrected

3) Figure 1: capitalize Immunoaffinity

Corrected

4) Line 544, refer to the lead authors names, or find another way to describe the studies. The way it is done is not proper. Please do the same for lines 610 and 613.

For Line 544, we moved the references a sentence early and rephrased it to clarify the message. For Line 610-613, we refer to the study once more to clarify the message.

5) Throughout the document there are areas in which microcystin is still written out and not abbreviated.

The initial instance of microcystins in line 38 was amended to include the abbreviation, and aside from table headers all other instances of ‘microcystin’ in the paper was replaced with ‘MCs’, with some minor grammatical changes to keep the language correct.

6) Cylindrospermopsin showing both CYL and CYN abbreviations. I think CYL is the most common.

All CYL instances were replaced with CYN after internal discussions of which nomenclature was preferred.

7) In the discussion, section 5.0, Prymnesins are not discussed. Why is that?

This omission was by accident, as the state of the science for these compounds is still uncertain. A paragraph addressing this has been added.

No Further Comments.

Thank you for your help with the review process. You've helped us significantly improve the manuscript. 

Reviewer 3 Report

No further comments.

Author Response

Thank you for your reviews on this paper. You've helped us significantly improve the review.